Effect of prognostic nutritional index on laboratory parameters and survival in metastatic colorectal cancer patients treated with fruquintinib: a retrospective study

Wang Zeng 1 2
Zhao Sining 3
Zhang Xuan 1
Mao Xinyi 1
Yang Guonong 1
Yuan Meiqin 4 yuanmq@zjcc.org.cn
Zhou Xiaofang 1 zhouxf@zjcc.org.cn
1 Department of Pharmacy, Zhejiang Cancer Hospital , Hangzhou, Zhejiang , China
2 Zhejiang Cancer Hospital, The Key Laboratory of Zhejiang Province for Aptamers and Theranostics, Hangzhou Institute of Medicine (HIM), Chinese Academy of Sciences , Hangzhou, Zhejiang , China
3 School of Pharmaceutical Sciences, Zhejiang Chinese Medical University , Hangzhou, Zhejiang , China
4 Department of Colorectal Medicine, Zhejiang Cancer Hospital , Hangzhou, Zhejiang , China
Oliveira Sonia
Electronic publication date: 2024 Nov 29
Publication date: 2024
Volume: 12
Electronic Location ID: e18565
Received 2024 Jul 23; Accepted 2024 Oct 31
Copyright: © 2024 Wang et al.
Copyright year: 2024
Copyright holder: Wang et al.
License: This is an open access article distributed under the terms of the Creative Commons Attribution License, which permits unrestricted use, distribution, reproduction and adaptation in any medium and for any purpose provided that it is properly attributed. For attribution, the original author(s), title, publication source (PeerJ) and either DOI or URL of the article must be cited.
License URL: https://creativecommons.org/licenses/by/4.0/

Keywords: Fruquintinib, Metastatic colorectal cancer, Prognostic nutritional index, Inspection indicators, Survival

Funding: 1022 Talent Training Program of Zhejiang Cancer Hospital, Zhejiang Provincial National Science Foundation of China LBY23H080003 Zhejiang Provincial Medical and Health General Research Program Project 2023KY564 Chinese Medicine Science and Technology Plan of Zhejiang Province 2021ZB214, 2022ZB052 This work was supported by the 1022 Talent Training Program of Zhejiang Cancer Hospital, Zhejiang Provincial National Science Foundation of China (No. LBY23H080003), the Zhejiang Provincial Medical and Health General Research Program Project (No. 2023KY564), the Chinese Medicine Science and Technology Plan of Zhejiang Province (No. 2021ZB214, 2022ZB052). The funders had no role in study design, data collection and analysis, decision to publish, or preparation of the manuscript.

==============================
Objective

Fruquintinib, a novel anti-angiogenic targeted drug, has gained widespread application in the treatment of metastatic colorectal cancer. This study aims to investigate the impact of the prognostic nutritional index (PNI) on the safety and survival outcomes of patients undergoing fruquintinib treatment for metastatic colorectal cancer.

Methods

A cohort of 106 patients with metastatic colorectal cancer, treated with fruquintinib at Zhejiang Cancer Hospital between 2019 and 2023, was included in this study. Clinical and laboratory data were subjected to chi-square and t-tests for analysis. PNI values were calculated using a specific formula. The optimal thresholds (cut-off values) for post-treatment PNI were determined through the ROC curve analysis. Kaplan-Meier analysis and the Log-rank test were employed to evaluate progression-free survival (PFS) and overall survival (OS) based on PNI. Multivariate Cox regression model was used to determine independent prognostic factors which influenced survival time.

Results

The study enrolled 106 colorectal cancer patients treated with fruquintinib. Stratified PNI analysis revealed significant differences in various indicators between high and low PNI groups after treatment with fruquintinib. Notably, after fruquintinib treatment, the high PNI group demonstrated elevated levels in white blood cells, lymphocytes, basophils, red blood cells, hemoglobin, platelets, total protein, and albumin compared to the low PNI group. The median OS for patients with high PNI values was 467 days, significantly longer than the 182 days observed for patients with low PNI values (P < 0.05). Cox regression analysis identified wild-type total RAS and BRAF, partial response (PR) + stable disease (SD), and high PNI values as influencing factors for OS in colorectal cancer patients. Additionally, PR + SD was an independent influencing factor for PFS in colorectal cancer patients (P < 0.05).

Conclusions

This study suggests that fuquinitinib can improve the survival of patients with metastatic colorectal cancer. Patients with high levels of PNI have a better prognosis and longer survival time, ensuring the nutritional status of patients can be a help to improve the treatment of fuquinitinib.

Introduction

Colorectal cancer (CRC) stands as one of the prevalent malignancies affecting the digestive tract in China. Over recent years, the incidence of CRC has exhibited a steady increase, with a concerning trend towards a younger demographic, posing a significant threat to human life and well-being. The early manifestations of colorectal cancer often lack clarity, and the onset is insidious, making it challenging to detect and frequently overlooked. By the time various clinical symptoms manifest, diagnosis typically occurs at intermediate to advanced stages, often accompanied by organ metastasis, thereby missing the opportune window for optimal surgical intervention. In the context of metastatic colorectal cancer (mCRC), the liver emerges as the primary target organ for hematogenous metastasis of colorectal cancer (Pretzsch et al., 2019). Liver metastasis stands out as a predominant contributor to mortality in colorectal cancer patients.

Tumors need to continuously induce angiogenesis to obtain sufficient nutrients and oxygen for growth and metastasis. Traditional chemotherapy methods cannot meet the needs of patients. At present, it has entered the era of molecular targeted drug therapy, providing a new treatment for mCRC patients (Li et al., 2023). Fruquintinib is an anti-angiogenic small molecule tyrosine kinase inhibitor (TKI) independently developed in China. It has been approved for the treatment of metastatic colorectal cancer in the Chinese Guidelines for the Diagnosis and Comprehensive Treatment of Colorectal Cancer Liver Metastasis (2023 Edition) (Ren et al., 2023). As a domestic original drug, fruquintinib has the characteristics of high selectivity, low off-target toxicity and good tolerance. It mainly acts on three main targets of the VEGFR kinase family: VEGFR1, VEGFR2 and VEGFR3, and can inhibit the activity of VEGFR kinase at the molecular level (Liu et al., 2022). Fruquintinib inhibits the proliferation, migration and tube formation of vascular endothelial cells. Oral administration can effectively block the growth of new blood vessels associated with tumors and exert anti-tumor effects (Li, Cai & Deng, 2021).

Cancer cells and immunity are like enemies in an offensive and defensive battle. When the human immunity is low, cancer cells will escape the surveillance of the immune system through camouflage, survive in the body and multiply, and eventually induce cancer. The prognostic nutritional index (PNI) was originally used to evaluate the nutritional and immune status of patients undergoing gastrointestinal surgery. In recent years, due to its simple and feasible characteristics in clinical application, it has also been predicted and verified to be an effective prognostic biomarker for a variety of malignant tumors, including gastric cancer, esophageal cancer, colorectal cancer, non-small cell lung cancer (Korkmaz et al., 2023; Qi et al., 2021; Kocak et al., 2023; Ryu et al., 2023). This study mainly focused on patients with metastatic colorectal cancer, to evaluate the effect of PNI on treatment and survival prognosis.

PNI is calculated based on serum lymphocyte count and serum albumin values. Lymphocytes play a crucial role in inhibiting the growth and proliferation of tumor cells, contributing significantly to the immune response. The decrease in the number of lymphocytes indicates that the immune function of the body is weakened, and cancer cells are more prone to immune escape, creating a favorable microenvironment for tumor recurrence and poor prognosis. Meanwhile, serum albumin, synthesized by the liver, serves as an indicator of the patient’s nutritional status. Low serum albumin levels are associated with an increased inflammatory response to tumors, reflecting poor nutritional status (Luvián-Morales et al., 2019). The combined PNI index derived from these two parameters offers an objective and effective assessment of both the nutritional status and immune level of patients. Early determination of the PNI proves beneficial for clinicians in formulating individualized treatment plans, holding immense significance for improving the survival outcomes of patients with colorectal cancer (Ucar et al., 2020). In conclusion, monitoring and optimizing the level of PNI can help improve the immune function of patients and reduce the level of inflammation, thereby prolonging the survival of patients and improving the quality of prognosis, which is particularly important in the treatment of metastatic colorectal cancer.

Therefore, this study retrospectively analyzed the clinical data of 106 patients with metastatic colorectal cancer treated with fruquintinib to explore the impact of the PNI on laboratory indicators and survival. The objective is to provide valuable practical insights for routine clinical practice.

Methods

Patients

A retrospective analysis was conducted on 264 patients diagnosed with metastatic colorectal cancer who sought treatment at Zhejiang Cancer Hospital between 2019 and 2023 and received fruquintinib treatment. Following the application of inclusion and exclusion criteria, 106 patients with metastatic colorectal cancer were selected for inclusion in this study (refer to Fig. 1), all of whom were aged 18 years or older. Patient follow-up was conducted through outpatient visits or telephone communication, extending until January 31, 2023. The study was approved by the ethics committee and institutional review board of Zhejiang Cancer Hospital and received a waiver of the need for informed consent from participants (IRB-2024-495).

Figure 1 Follow-up flow chart of patient information collection.

Study flow chart out of the 264 patients who were treated with fruquintinib, 158 patients were excluded: 151 because their inspection data is incomplete, seven because their follow-up records of OS data are missing.

Inclusion criteria for colorectal cancer

1) Age 18–80 years

2) Colorectal cancer diagnosed by pathology or cytology

3) At least one measurable number of metastatic lesions

4) Expected survival time exceeding 12 weeks

5) Complete blood routine and biochemical data after treatment

Excluded criteria

1) Failed to complete treatment and follow-up on time

2) Missing laboratory or survival data

3) Active infection

4) Chronic inflammatory disease

5) Receive albumin replace treatment

6) Long term or high-dose steroids therapy

Clinical data collection

Collected through the electronic medical record system of Zhejiang Cancer Hospital:

(1) Collect basic data of patients: including name, hospitalization number, gender, age, etc.

(2) Clinicopathological data: KRAS, NRAS, BRAF gene status, primary site, primary tumor resection, metastatic site, number of treatment lines, start time and end time of medication, combined use of PD-1, efficacy evaluation, duration of treatment, blood routine index, biochemical index, NK complete index and nutritional score after treatment with fruquintinib.

The survival status of patients was assessed using two key indicators: overall survival (OS) and progression-free survival (PFS). Information on survival, overall survival, and progression-free survival was gathered through follow-up procedures.

Statistics

Statistical analysis was performed using SPSS 26.0 software, and a significance level of P < 0.05 was considered statistically significant.

The 106 patients were categorized into different subgroups, and the chi-square test was employed to compare variations in the clinical characteristics among patients. Descriptive statistics, including mean (AV ± SD) for normally distributed variables and median (Min, Max) for non-normally distributed variables, were used to depict the distribution characteristics of laboratory indexes after fruquintinib treatment. The PNI was calculated using the formula PNI = serum albumin (g/L) + 5 × total number of peripheral blood lymphocytes (×109/L). The ROC curve was utilized to compare PNI values after treatment, determining the optimal cut-off value for PNI based on patient survival.

OS was defined as the duration from the initiation of fruquintinib treatment to death from any cause, while PFS was defined as the interval from fruquintinib initiation to disease progression or death. Kaplan-Meier analysis generated survival curves, and differences were compared using the Log-rank test. Cox regression analysis was employed to calculate the hazard ratio (HR) and statistical index (P value). Factors with P < 0.1 in univariate analysis were selected for further determination of prognostic factors, considering a significance level of P < 0.05.

The cut-off value, defining the critical point for classifying PNI after fruquintinib treatment based on patient survival, was determined using the ROC curve. Patients were grouped into high and low levels based on these cut-off values. The Yoden index, calculated using the formula: Yoden index = sensitivity + specificity − 1, determined the optimal diagnostic threshold corresponding to the maximum Yoden index. The cut-off value for PNI after treatment was 45.8 (Fig. 2).

Figure 2 ROC curve of post-treatment PNI.

Results

Basic information about the patient

Between 2019 and 2023, a total of 264 patients underwent fruquintinib treatment. Out of these, 158 patients were excluded from the analysis, and the data of the remaining 106 patients were subjected to detailed examination.

The study encompassed 106 patients diagnosed with colorectal cancer, comprising 70 males and 36 females, with ages ranging from 28 to 78 years. Within the cohort, 28 patients were aged 65 years or older, while 78 patients were below the age of 65. The genetic profile revealed 41 cases of total RAS gene mutation, 32 wild-type cases, and 33 cases without detection. Additionally, BRAF mutation was identified in two cases, while 70 cases exhibited a wild-type BRAF, and 34 cases were not detected. Tumor distribution showed 48 patients with primary tumors in the colon, 58 in the rectum, 82 in the left colon, and 24 in the right colon. Regarding metastasis, 11 cases had one tumor site affected, 45 cases had two sites, and 50 cases had three sites. In terms of chemotherapy lines, four cases received below the third line, 78 cases received the third line, and 24 cases received the fourth line and above. Notably, 44 patients were combined with PD-1, while 62 patients were not treated with it. The curative effect score reflected 32 cases of PD, one case of PR, and 73 cases of SD. The study observed 86 deaths and 20 survivors (Table 1).

Table 1 Clinical data of 106 patients.

		All patients (n = 106)	
Gender			
	Male	70 (66.04)	
	Female	36 (33.96)	
Age (years)			
	AV ± SD	57.93 ± 9.76	
	≥65	28 (26.42)	
	<65	78 (73.58)	
Total RAS			
	Mutation	41 (38.68)	
	Wild	32 (30.19)	
	Unidentified	33 (31.13)	
BRAF			
	Mutation	2 (1.89)	
	Wild	70 (66.04)	
	Unidentified	34 (32.08)	
Tumor site			
	Colon	48 (45.28)	
	Rectum	58 (54.72)	
	Left colon	82 (77.36)	
	Right colon	24 (22.64)	
Metastatic site			
	One	11 (10.38)	
	Two	45 (42.45)	
	Three	50 (47.17)	
Number of treatment lines			
	Below the three lines	4 (3.77)	
	Three lines	78 (73.59)	
	More than three lines	24 (22.64)	
PD-1			
	Combined	44 (41.51)	
	Not combined	62 (58.49)	
Efficacy			
	PD	32 (30.19)	
	PR	1 (0.94)	
	SD	73 (68.87)	
Death or not			
	Yes	86 (81.13)	
	No	20 (18.87)	
Note:

Data are shown as median (range) or n (%).

Patient medication safety analysis

Statistical table of test data before and after treatment

The results showed that the red blood cell count in the blood routine index increased (P = 0.015 < 0.05), with statistical difference. The albumin in the biochemical index decreased (P = 0.001 < 0.01) and the L-r-glutamyltransferase increased (P = 0.003 < 0.01) after the treatment of fruquintinib, with significant statistical differences (Table 2).

Table 2 Comparison and analysis based on stratified test data before and after treatment.

Index		Number of cases before treatment (n = 106)	Number of cases after treatment (n = 106)	P	
AV ± SD/median (MIN, MAX)	AV ± SD/median (MIN, MAX)	
Blood routine	Leucocyte	5.84 ± 2.17	6.33 ± 2.25	0.104	
Neutrophil count	3.95 ± 1.81	4.40 ± 1.82	0.070	
Lymphocyte count	1.21 ± 0.54	1.29 ± 0.65	0.303	
Monocyte count	0.47 ± 0.24	0.43 ± 0.19	0.135	
Eosinophil count	0.1 (0, 1.4)	0.1 (0, 1.5)	0.796	
Basophil count	0.02 ± 0.01	0.03 ± 0.02	0.209	
Red blood cell count	3.95 ± 0.51	4.15 ± 0.67	0.015*	
Hemoglobin	118.26 ± 22.93	121.15 ± 26.00	0.387	
Platelet count	188.58 ± 75.58	188.26 ± 88.24	0.977	
Biochemistry	Total protein	68.57 ± 7.30	67.15 ± 7.45	0.158	
Albumin	38.81 ± 4.89	36.67 ± 4.86	0.001**	
Aspartate aminotransferase	35.95 ± 24.68	37.19 ± 22.33	0.700	
L-r-glutamyltransferase	78.5 (4, 680)	89 (14, 4,121)	0.003**	
Creatine kinase	96.58 ± 68.43	74 (23, 877)	0.559	
Total bilirubin	11.55 ± 6.03	11.18 ± 5.54	0.642	
Direct bilirubin	3.84 ± 2.54	4.04 ± 3.29	0.618	
Indirect bilirubin	7.71 ± 3.94	7.14 ± 3.19	0.246	
Total bile acid	6.25 (1.1, 104.8)	6.5 (0.7, 102.5)	0.988	
NK complete set	Total T cells	69.35 ± 12.53	67.27 ± 13.65	0.513	
Helper effector T cells	38.69 ± 10.22	38.26 ± 12.19	0.873	
Inhibition of cytotoxic T cells	25.88 ± 10.54	25.05 ± 11.03	0.753	
CD4/CD8 ratios	1.79 ± 1.02	1.73 ± 0.89	0.791	
Nutrition score	1.75 ± 1.06	1.88 ± 1.25	0.399	
Notes:

* P < 0.05, with statistical difference.

** P < 0.01, with significant statistical difference.

Statistical table of PNI subgroup test data after treatment

The PNI cut-off value of 45.8 was utilized as the categorical variable for grouping statistics. The analysis revealed notable differences between the high PNI group and the low PNI group after fruquintinib treatment. Specifically, there were increases in white blood cell count (P = 0.002 < 0.01), lymphocyte count (P < 0.001), basophil count (P = 0.012 < 0.05), red blood cell count (P < 0.001), hemoglobin (P < 0.001), and platelet count (P = 0.048 < 0.05) within the blood routine indexes. Additionally, the total protein in biochemical indicators exhibited an increase (P < 0.001), as did albumin (P < 0.001). These differences were statistically significant (Table 3).

Table 3 Comparison and analysis based on stratified test data of PNI CUT-OFF values after treatment with fruquintinib.

Index	Post-treatment PNI	≥Cut-off (n = 36)	<Cut-off (n = 70)	P	
AV ± SD/median (MIN, MAX)	AV ± SD/median (MIN, MAX)	
Blood routine	Leucocyte	7.25 ± 2.46	5.85 ± 1.99	0.002**	
Neutrophil count	4.77 ± 1.94	4.21 ± 1.75	0.133	
Lymphocyte count	1.75 ± 0.71	1.06 ± 0.47	<0.001***	
Monocyte count	0.48 ± 0.20	0.41 ± 0.19	0.083	
Eosinophil count	0.1 (0, 1.5)	0.1 (0, 0.7)	0.272	
Basophil count	0.03 ± 0.02	0.02 ± 0.01	0.012*	
Red blood cell count	4.47 ± 0.54	3.98 ± 0.67	<0.001***	
Hemoglobin	133.72 ± 16.01	114.69 ± 27.82	<0.001***	
Platelet count	211.86 ± 80.43	176.13 ± 90.16	0.048*	
Biochemistry	Total protein	71.21 ± 5.16	65.07 ± 7.61	<0.001***	
Albumin	40.91 ± 3.20	34.49 ± 4.07	<0.001***	
Aspartate aminotransferase	32.47 ± 14.90	39.61 ± 25.07	0.119	
L-r-glutamyltransferase	105.97 ± 88.31	89.5 (16, 4,121)	0.465	
Creatine kinase	103.33 ± 60.05	65 (24,877)	0.160	
Total bilirubin	10.53 ± 3.47	11.52 ± 6.35	0.385	
Direct bilirubin	3.18 ± 1.29	4.49 ± 3.88	0.052	
Indirect bilirubin	7.35 ± 2.59	7.03 ± 3.47	0.631	
Total bile acid	6.1 (1.3, 102.5)	7.15 (0.7, 56.5)	0.258	
NK complete set	Total T cells	65.23 ± 13.88	69.01 ± 13.72	0.492	
Helper effector T cells	36.28 ± 6.63	39.96 ± 15.55	0.455	
Inhibition of cytotoxic T cells	25.04 ± 9.59	25.06 ± 12.50	0.997	
CD4/CD8 ratios	1.67 ± 0.73	1.78 ± 1.03	0.755	
Nutrition score	1.74 ± 1.07	1.95 ± 1.34	0.421	
Notes:

* P < 0.05, with statistical difference.

** P < 0.01, with significant statistical difference.

*** P < 0.001, with extremely significant statistical differences.

The cut-off value of PNI after treatment was 45.8.

COX regression analysis

OS

COX regression analysis was utilized to examine the clinical characteristics of the 106 colorectal cancer patients and the impact of PNI after fruquintinib treatment on OS. In univariate regression analysis, total RAS (P = 0.083), BRAF (P = 0.001), efficacy (P = 0.004), and post-treatment PNI (P = 0.004) emerged as influencing factors of OS (P < 0.1). Further multivariate analysis of the aforementioned four variables indicated that total RAS mutation had a statistically significant effect on OS (HR = 0.622, 95% CI [0.398–0.970], P = 0.036). Additionally, BRAF mutation demonstrated a statistically significant impact on OS (HR = 0.053, 95% CI [0.011–0.255], P < 0.001). Patients with PD exhibited a statistically significant effect on OS compared to those with PR and SD (HR = 0.462, 95% CI [0.290–0.738], P = 0.001). Post-treatment PNI higher than 45.8 was associated with a statistically significant effect on OS (HR = 2.081, 95% CI [1.267–3.420], P = 0.004), while other variables did not exhibit a statistically significant impact on OS (P > 0.05). The results indicated that in patients with wild-type total RAS and BRAF mutations, those with high PNI values and PR + SD efficacy after fruquintinib treatment experienced longer overall survival and demonstrated a better prognosis (Table 4).

Table 4 Univariate and multivariate analyses of OS in post-treatment PNI.

Post-treatment PNI—OS	Univariate analysis	Multivariate analysis	
HR (95% Cl)	P	HR (95% Cl)	P	
Gender (male vs. female)	0.960 [0.579–1.592]	0.875			
Age (≥65 vs. <65)	1.519 [0.905–2.550]	0.113			
Total RAS (mutation vs. not mutation)	0.642 [0.389–1.059]	0.083*	0.622 [0.398–0.970]	0.036*	
BRAF (mutation vs. not mutation)	0.059 [0.011–0.307]	0.001*	0.053 [0.011–0.255]	<0.001*	
Tumor site (colon vs. rectum)	1.241 [0.691–2.228]	0.470			
Left and right colon (left colon vs. right colon)	1.369 [0.708–2.648]	0.351			
Metastatic site (<2 vs. ≥2)	0.809 [0.364–1.802]	0.605			
Number of treatment lines (≤3 vs. >3)	1.024 [0.582–1.801]	0.935			
PD-1 (Not combined vs. combined)	1.111 [0.681–1.814]	0.673			
Efficacy (PD vs. PR + SD)	0.460 [0.271–0.783]	0.004*	0.462 [0.290–0.738]	0.001*	
Post-treatment PNI (≥cut-off vs. <cut-off)	2.216 [1.284–3.825]	0.004*	2.081 [1.267–3.420]	0.004*	
Notes:

* P < 0.05, with statistical difference.

The cut-off value of PNI after treatment was 45.8.

PFS

COX regression analysis was employed to scrutinize the clinical characteristics of 106 colorectal cancer patients and the impact of PNI after fruquintinib treatment on PFS. Univariate regression analysis revealed that efficacy (P < 0.001) served as the influencing factor of PFS (P < 0.1). Further multivariate analysis unveiled that the effect of PD on PFS was statistically significant compared to PR and SD (HR = 0.192, 95% CI [0.106–0.349], P < 0.001). However, the impact of other variables on PFS was not statistically significant (P > 0.05). The findings indicated that patients with PR + SD after fruquintinib treatment experienced longer progression-free survival and demonstrated a better prognosis (Table 5).

Table 5 Univariate and multivariate analyses of PFS in post-treatment PNI.

Post-treatment PNI—PFS	Univariate analysis	Multivariate analysis	
HR (95% Cl)	P	HR (95% Cl)	P	
Gender (male vs. female)	1.180 [0.721–1.930]	0.510			
Age (≥65 vs. <65)	1.216 [0.726–2.034]	0.457			
Total RAS (mutation vs. not mutation)	0.691 [0.426–1.121]	0.134			
BRAF (mutation vs. not mutation)	0.306 [0.066–1.415]	0.129			
Tumor site (colon vs. rectum)	0.820 [0.453–1.485]	0.513			
Left and Right Colon (left colon vs. right colon)	0.997 [0.489–2.033]	0.993			
Metastatic Site (<2 vs. ≥2)	1.176 [0.537–2.575]	0.685			
Number of Treatment Lines (≤3 vs. >3)	0.931 [0.543–1.594]	0.793			
PD-1 (combined vs. Not combined)	1.319 [0.800–2.175]	0.278			
Efficacy (PD vs. PR + SD)	0.235 [0.126–0.440]	<0.001*	0.192 [0.106–0.349]	<0.001*	
Post-treatment PNI (≥cut-off vs. <cut-off)	1.449 [0.855–2.457]	0.168			
Notes:

* P < 0.05, with statistical difference.

The cut-off value of PNI after treatment was 45.8.

Survival analysis

OS

The analysis encompassed the basic information indices and survival time of all patients. Results indicated that the median survival time for the wild type in total RAS was 402 days longer than that of the other two groups, exhibiting statistical significance (P = 0.024 < 0.05). Furthermore, the median survival time for BRAF wild type was 257 days longer than that of the other two groups, displaying an extremely significant statistical difference (P < 0.001). The median survival time for the PD group was 174 days shorter than that of the PR + SD group, which was 355 days, and this difference was statistically significant (P = 0.002 < 0.01). Additionally, the median survival time for the high PNI value after treatment was 467 days longer than that of the low PNI value group, which was 182 days. This difference was extremely statistically significant (P < 0.001) (Table 6, Fig. 3).

Table 6 Median and P value of OS.

Index	Median survival time (days)	Log rank	
Gender	Male	260	0.338	
Female	244	
Age	≥65	384	0.407	
<65	243	
Total RAS	Mutation	182	0.024*	
Wild	402	
Unidentified	315	
BRAF	Mutation	48	<0.001***	
Wild	257	
Unidentified	249	
Tumor site	Colon	315	0.428	
Rectum	245	
Left and right colon	Left colon	260	0.330	
Right colon	244	
Metastatic site	One	402	0.432	
Two	246	
Three	245	
Number of treatment lines	Below the three lines		0.122	
Three lines	246	
More than three lines	245	
PD-1	Combined	315	0.148	
Not combined	245	
Efficacy	PD	174	0.002**	
PR + SD	355	
Post-treatment PNI	≥cut-off	467	<0.001***	
<cut-off	182	
Notes:

* P < 0.05, with statistical difference.

** P < 0.01, with significant statistical difference.

*** P < 0.001, with extremely significant statistical differences.

The cut-off value of PNI after treatment was 45.8.

Figure 3 Figure of OS difference.

PFS

The analysis encompassed the basic information indicators and survival time of all patients. Results revealed that the median survival time for men was 72 days longer than that of women, amounting to 56 days, with statistical significance (P = 0.048 < 0.05). Moreover, the median survival time for the wild type in total RAS was 112 days longer than that of the other two groups, exhibiting significant statistical difference (P = 0.003 < 0.01). Conversely, the median survival time for the BRAF mutant was 28 days shorter than that of the other two groups, displaying statistical significance (P = 0.037 < 0.05). The median survival time for the PD-1 combined group was 72 days longer than that of the non-combined group, amounting to 56 days, with significant statistical difference (P = 0.002 < 0.01). Furthermore, the median survival time for the PD group was 28 days shorter than that of the PR + SD group, which was 72 days, and this difference was extremely statistically significant (P < 0.001). After treatment, the median survival time for the high PNI value group and the low PNI value group was 72 days, with statistical significance (P = 0.041 < 0.05) (Table 7, Fig. 4).

Table 7 Median and P value of PFS.

Index	Median survival time (days)	Log rank	
Gender	Male	72	0.048*	
Female	56	
Age	≥65	72	0.603	
<65	72	
Total RAS	Mutation	56	0.003**	
Wild	112	
Unidentified	72	
BRAF	Mutation	28	0.037*	
Wild	72	
Unidentified	72	
Tumor site	Colon	61	0.632	
Rectum	72	
Left and right colon	Left colon	72	0.246	
Right colon	72	
Metastatic site	One	112	0.092	
Two	61	
Three	72	
Number of treatment lines	Below the three lines		0.068	
Three lines	72	
More than three lines	72	
PD-1	Combined	72	0.002**	
Not combined	56	
Efficacy	PD	28	<0.001***	
PR + SD	72	
Post-treatment PNI	≥cut-off	72	0.041*	
<cut-off	72	
Notes:

* P < 0.05, with statistical difference.

** P < 0.01, with significant statistical difference.

*** P < 0.001, with extremely significant statistical differences.

The cut-off value of PNI after treatment was 45.8.

Figure 4 Figure of PFS difference.

Discussion

The incidence and mortality rates of colorectal cancer are increasing, with the liver being the primary site of metastasis and recurrence in colon cancer. The advent of targeted therapy aims to exploit specific molecular vulnerabilities in cancer cells while sparing normal tissues, thereby minimizing side effects and improving patient prognosis (Huang et al., 2023). This article further affirms the substantial efficacy and safety of fruquintinib in treatment, and PNI is highlighted for its simple calculation and easy accessibility. PNI not only allows for the quantification of patients’ nutritional status but also serves as a reflection of cancer-related immune-inflammatory response (Hayama et al., 2022). In this study, the impact of PNI on the clinical characteristics, laboratory indicators, and survival of patients with colorectal cancer was thoroughly investigated.

This study retrospectively analyzed the clinical characteristics of 106 patients with colorectal cancer and found that patients with RAS wild-type had a higher survival rate than those with RAS mutations. Moreover, patients with elevated PNI demonstrated a relatively favorable prognosis. Notably, the PD-1 combination group exhibited a low proportion of total RAS gene mutations, resulting in a lower mortality rate than individuals who did not receive PD-1 in combination.

The RAS gene stands out as the most frequently mutated gene in human cancer, imparting substantial significance to its mutation status for the therapeutic efficacy of targeted drugs. KRAS and NRAS, two GTPase proteins encoded by RAS family member genes, play crucial roles in cancer development. Patients with KRAS gene mutations exhibit close associations with disease status, survival, and prognosis (Dai et al., 2020). In the present study, patients manifested simultaneous mutations in both total RAS and BRAF genes. The proportion of patients with combined total RAS mutations and PD-1 treatment was notably low, with a higher number of deaths compared to RAS wild-type patients, aligning with existing research findings (Levin-Sparenberg et al., 2020). Routine detection of clinical gene types in colorectal cancer patients is strongly recommended to inform personalized treatment strategies.

The study identified a correlation between the PNI value, patient age, and survival outcomes. This observation indicates that patients with a higher PNI value exhibit a relatively favorable prognosis and an extended survival time. Prior studies have emphasized that PNI serves not only as a prognostic indicator for colorectal cancer patients but also reflects their immune status and nutritional well-being to some extent (Xie et al., 2022). As an indicator of immunonutritional status, PNI can be easily applied in patients with metastatic colorectal cancer and has prognostic value (Keskinkilic et al., 2024). Patients with lower PNI may experience compromised body resistance, impeding the ability to curb tumor growth and diffusion. Implementing more effective nutritional interventions is recommended, as these interventions can positively impact prognosis. Therefore, assessing PNI values holds clinical significance in enhancing the overall quality of life for patients (Zhu et al., 2021).

The burgeoning advancements in molecular diagnosis and gene detection have paved the way for enhanced prognostic outcomes for patients. The integration of immunotherapy with targeted drug therapy has demonstrated substantial improvements in the therapeutic efficacy against solid tumors (Hansen, Qvortrup & Pfeiffer, 2021). The data from this study indicated a correlation between the use of PD-1, total RAS status, treatment efficacy, and overall survival. Within the PD-1 combination group, there was a low incidence of total RAS gene mutations (25%). Patients in this group exhibited a more stable condition, predominantly achieving a disease stabilization (SD) response (84.1%), and demonstrated a lower mortality rate compared to the non-combined PD-1 group (70.5%). This observation suggests that PD-1 plays a pivotal role in modulating tumor immunity and inflammatory responses.

Moreover, the study findings align with those of Wang et al. (2020), supporting the notion that the combined treatment approach with fruquintinib offers superior overall performance. Tumor blood vessels possess unique biological characteristics, and PD-1 is primarily expressed on the surface of activated T cells. When coupled with anti-angiogenic drugs, this combination can transform the immune microenvironment, regulating the migration of T cells to the tumor microenvironment and mitigating subsequent immune responses (Li et al., 2022). The synergistic effect of this dual approach enhances tumor inhibition, effectively boosting the anticancer efficacy of fruquintinib and prolonging overall survival. This approach empowers patients to attain efficient resistance against cancer.

Furher, the receiver operating characteristic (ROC) curve analysis was used to select the most appropriate cut-off point for PNI to distinguish patients at high risk of death. These colorectal cancer patients were stratified into high and low PNI groups based on the critical values of PNI after treatment (45.8). Post-treatment, the differences persisted, showing increased counts in various test indexes for the high PNI group, including white blood cells, lymphocytes, basophils, red blood cells, hemoglobin, platelets, total protein, and albumin. This suggests that patients with a high PNI value exhibit higher examination indexes compared to those with a low PNI value.

Following fruquintinib treatment, colorectal cancer patients with high PNI levels exhibited statistically higher levels of basophils, red blood cells, platelets, and total protein compared to patients with low PNI levels. Basophils, a type of white blood cells, play a crucial role in immune response and allergic reactions, possessing phagocytic and anti-inflammatory effects. Reduced basophil count is associated with low immunity, potentially leading to a poor prognosis. Red blood cells primarily transport oxygen and carbon dioxide, and a decrease in their numbers can result in reduced oxygen-carrying capacity and the development of anemia symptoms. Given that colorectal cancer patients often experience hematochezia and chronic gastrointestinal bleeding, leading to anemia, close attention to this condition is warranted in clinical practice (Zhou et al., 2023). Platelets contribute to hemostasis and capillary nourishment. Monitoring platelet count dynamically can aid in assessing the clinical treatment effect and prognosis of tumors, serving as a potential reference index (Herold et al., 2022). Low total protein levels indicate abnormal nutritional status and impaired liver synthesis function, which may be accompanied by body edema and infection. In summary, a high PNI value appears to positively influence the patient’s blood routine indicators and nutritional parameters.

This study further analyzed the survival data of patients, and the total RAS, BRAF, efficacy, and PNI values were the influencing factors of OS. Efficacy is an independent prognostic factor for PFS. Furthermore, multivariate Cox regression analysis demonstrated that a high PNI value stood out as a prognostic factor significantly associated with extended OS in patients, leading to a substantial increase in survival time. Patients with a high PNI exhibited a median OS of 467 days, which was 2.5 times longer than those with a low PNI, and this difference achieved statistical significance. These findings align with the research outcomes reported by Hu et al. (2020).

However, it is noteworthy that the PNI value did not exhibit a significant correlation with PFS. As PNI is intricately linked with serum albumin and peripheral blood lymphocyte count, it serves as a reflection of the delicate balance between inflammation and immune response within the tumor microenvironment. Utilizing the PNI value for predicting the survival of patients with colorectal cancer can, therefore, contribute significantly to enhancing the consideration of nutritional status in clinical decision-making (Kim et al., 2023).

Studies have shown that PR + SD is an independent factor affecting PFS in patients with colorectal cancer. Notably, patients exhibiting PR and SD after treatment displayed a mOS of 355 days, along with a mPFS of 72 days—more than twice the duration observed in patients experiencing disease progression (PD). The risk of mortality in patients with PR + SD demonstrated a substantial 80% reduction compared to those with PD, emphasizing a longer progression-free survival period and a more favorable prognosis. This study reaffirms the significant efficacy of fruquintinib in mCRC. A comprehensive evaluation of efficacy in clinical practice proves instrumental in guiding subsequent treatment decisions for colorectal cancer.

However, this study has some limitations. Due to this being a retrospective study based on a single center, after screening for inclusion and exclusion criteria, the actual number of cases used for statistical analysis was 106, which is relatively insufficient in sample size. Therefore, further validation of our findings is needed in large sample, prospective, multicenter clinical trials.

Conclusion and foresight

In this investigation, various factors, including PNI levels and treatment efficacy, exhibited significant correlations with the clinical outcomes of patients with metastatic colorectal cancer. Following fruquintinib treatment, patients with colorectal cancer and high PNI levels demonstrated elevated levels of white blood cells, lymphocytes, basophils, red blood cells, hemoglobin, platelets, total protein, and albumin, indicative of a pronounced therapeutic effect and extended OS time. Consequently, the survival analysis results underscore the role of PNI value as an influencing factor for OS, establishing a close association with long-term prognosis.

In summary, PNI emerges as a positive factor impacting laboratory indicators and prognostic survival among patients with metastatic colorectal cancer treated with fruquintinib. PNI evaluation proves valuable in predicting the risk of clinical complications and the prognostic quality of life. Nonetheless, it’s essential to acknowledge certain limitations in this study, such as the relatively small sample size and limited follow-up duration. Anticipating results from future prospective multi-center studies with larger sample sizes would provide further insights.

Supplemental Information

Supplemental Information 1 Comparison of Diagnostic Methods.

Supplemental Information 2 Survival Data Table.

Additional Information and Declarations

Competing Interests

Author Contributions

Human Ethics

Data Availability

The authors declare that they have no competing interests.

Zeng Wang conceived and designed the experiments, authored or reviewed drafts of the article, and approved the final draft.

Sining Zhao performed the experiments, authored or reviewed drafts of the article, and approved the final draft.

Xuan Zhang analyzed the data, prepared figures and/or tables, and approved the final draft.

Xinyi Mao analyzed the data, prepared figures and/or tables, and approved the final draft.

Guonong Yang analyzed the data, prepared figures and/or tables, and approved the final draft.

Meiqin Yuan conceived and designed the experiments, authored or reviewed drafts of the article, and approved the final draft.

Xiaofang Zhou conceived and designed the experiments, authored or reviewed drafts of the article, and approved the final draft.

The following information was supplied relating to ethical approvals (i.e., approving body and any reference numbers):

The study was approved by the ethics committee and institutional review board of Zhejiang Cancer Hospital (IRB-2024-495).

The following information was supplied regarding data availability:

The figures illustrate the patient screening process, the optimal PNI cut-off values post-treatment, and the differences in progression-free survival (PFS) and overall survival (OS) based on PNI levels. These data are used to analyze the impact of PNI levels on patient survival.

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
