# Peer review of "Effect of prognostic nutritional index on laboratory parameters and survival in metastatic colorectal cancer patients treated with fruquintinib: a retrospective study"

_PeerJ, doi:10.7717/peerj.18565_

## Round 0.1 · original submission · Minor Revisions

Dear authors, thank you for your submission. Your manuscript requires a few revisions before acceptance for publication. Please, refer to the reviewers' comments for further details.

·

Basic reporting

First of all, I congratulate you for this study.

The article is generally well written, but there are a few points that need to be edited in the study.

Those are;

1- References should be at the end of each sentence.

2- In the abstract section, since the abbreviation PR+SD is mentioned there for the first time, the open form should be stated.

3- In the introduction, the relationship between immune status and cancer and why PNI, one of these markers, is important in colorectal cancer should be briefly mentioned.

Experimental design

Some corrections are needed in the material method section. In the material method section, since the albumin and lymphocyte values ​​are used in the PNI calculation, it should be stated whether the patients use steroids, have chronic inflammatory disease, receive albumin replacement, and have active infection.

Validity of the findings

It would be beneficial for you to make the corrections I have mentioned below in the findings. Those are;
1- In the PFS section, it would be appropriate to state the actual p value instead of p = 0.
2-The location of the tumor in the colon should be stated in medical terms as right colon and left colon (instead of right half and left half).

Some additions should be made in the discussion section.
1- The limitations of the study should also be mentioned.
2- More recent publications on PNI in metastatic colorectal cancer should be included in the discussion and an addition should be made regarding the PNI cut-off value (DOI: 10.1007/s00520-024-08572-6)
3- It would be more appropriate to organize the discussion as a whole rather than dividing it into sections; some parts of the first part of the discussion seem to be repeated.

·

Basic reporting

The study is well-executed, with significant clinical relevance. The analysis of the impact of PNI on survival outcomes provides practical insights for clinicians treating mCRC.
The manuscript provides valuable insights into how nutritional status, as measured by PNI, can influence survival in mCRC patients, particularly those treated with fruquintinib. However, PNI has been used as a survival marker not only in colon cancers but also in GI cancers (https://doi.org/10.1007/s12029-023-00972-x). In addition, PNI variability predicted survival in metastatic colon cancer (https://doi.org/10.1007/s00520-023-07829-w). In this sense, the introduction section can be expanded.
The small sample size (106 patients) limits the generalizability of the results.

Experimental design

The study addresses a meaningful gap by investigating the role of PNI in the prognosis of mCRC patients treated with fruquintinib, a newer anti-angiogenic drug. The research is within the journal's scope, with a clear definition of objectives.
The methodology is well-detailed, allowing replication.

Validity of the findings

The statistical methods employed are appropriate and sound. Kaplan-Meier survival analysis, Cox regression, and the ROC curve are well-suited for the study's objectives. All underlying data is provided, and the findings are well-supported by the data.
The conclusions drawn from the data are clear and are aligned with the original research question.

Additional comments

Overall, this manuscript meets the standards for publication. I recommend minor revisions to improve the clarity of the introduction, and discussios to strengthen the study's impact.

---

## Round 0.2 · accepted · Accept

Dear authors, once again thank you for your submission and work. I am now accepting your work for publication.

·

Basic reporting

The recommendations and regulations for this section have been implemented.

Experimental design

The recommendations and regulations for this section have been implemented.

Validity of the findings

The recommendations and regulations for this section have been implemented.

·

Basic reporting

No comments

Experimental design

No comments

Validity of the findings

No comments

Additional comments

Revisions are sufficient